# The Disclosures of Information on Cybersecurity in Listed Companies in Latin America—Proposal for a Cybersecurity Disclosure Index

**Maricela Ramírez** [1,*] , **Lázaro Rodríguez Ariza** [2] , **María Elena Gómez Miranda** [2] **and Vartika** [3]

1 Faculty of Economic and Administrative Sciences, Pedagogical and Technological University of Colombia, Tunja 150001, Colombia
2 Department of Financial Economics and Accounting, University of Granada, 18071 Granada, Spain; lazaro@ugr.es (L.R.A.); melena@ugr.es (M.E.G.M.)
3 Indian Institute of Management Rohtak, Rohtak 124010, India; vartikajnu3007@gmail.com
* Correspondence: maricela@correo.ugr.es; Tel.: +57-3204878826

**Abstract:** For the corporate sphere, cybersecurity becomes an inescapable business responsibility, and accountability becomes a way of providing trust and ensuring resilience against cyber risks and high-impact cyber threats. The purpose of this study was to create a disclosure index that allows analysis of the scope of the disclosure of voluntary and mandatory cybersecurity information. The content analysis technique used focuses on the examination and identification of the cybersecurity information revealed in the annual reports and the 20 F annual forms of the companies with the highest stock market prices in Argentina, Brazil, Chile, Colombia, Mexico, and Peru during the period of 2016–2020. Longitudinal analysis indicates an increase over time in the disclosures and scope of information. The findings highlight that the country with the highest related disclosure is Argentina; the most extensive disclosures are due to the financial sector; and the strategy dimension represents the greatest weight in the index score. The study provides a novel instrument for measuring the content of disclosure on cybersecurity that is applicable in any specific context. In this case, the scope of disclosure in Latin America—a region which, according to our research, does not have previous studies on the subject—is evaluated.

**Keywords:** cybersecurity disclosure; disclosure index; cybersecurity governance; cybersecurity strategies; cybersecurity risk management; financial implications of cybersecurity risk

## 1. Introduction

The World Economic Forum [1] presents among the emerging risks of high probability the cybersecurity failures generated by the rapid increase in digitization. In this regard, professionals from the insurance industry across six regions of the planet, including Latin America, pointed out in the Axa survey [2] that the second-most important emerging risk—after that of climate change—comes due to the failures of cybersecurity, a perception that has almost doubled in importance since 2019, with a rate of 54%. Cybersecurity has become a major problem faced by most organizations [3], and in a digitally-connected world, it presents ongoing risks and threats to capital markets and companies operating in all industries [4]. According to King [5], technology is now the source both of many future opportunities for an organization, as well as of potential disruptions, and is an excellent example of how risk and opportunity are increasingly becoming two sides of the same coin. Now, returning to the definition of corporate social responsibility ISO 26,000 [6], the relationship between it and the measurement of the organizational impact is noted, and is understood as that positive or negative change that is generated in society, the economy or environment produced—in whole or in part—by the past and present decisions and activities of an organization. According to Rashid et al. [7], cybersecurity should be thought

of as a clear set of processes that reduce the risk of harm to people and the company. Thus, accountability becomes a way of communicating—based on financial and non-financial information—the effects of the organization's actions with society. In this framework, disclosure on cyber risk management is a practice that increases significantly in corporate annual reports and voluntary and mandatory reports, as well as on company websites. In this regard, companies should strengthen their cybersecurity disclosures in order to demonstrate responsibility and commitment to this issue and build stakeholder confidence in how it is being prioritized, managed, and monitored as a critical business threat and strategic opportunity [8].

In this context, Latin American companies have been gradually incorporating disclosures on cybersecurity in their corporate annual reports on a voluntary basis based on risk management, among which it is increasingly common to find cybersecurity risk and/or information security. In some cases, this is initially revealed as an emerging risk, and in subsequent periods as an operational risk. Although there is no exclusive regulation in the region for the disclosure of cybersecurity information in the annual report, some regulations issued by the surveillance and control entities regarding corporate governance, risk management and data privacy have generated an increased disclosure on cybersecurity—especially in the financial sector. At the same time, Latin American companies that participate as issuers in the US stock market also report the risk factors related to cyber risk on a mandatory basis in the annual report 20 F item 3D, according to SEC [4].

For the international stock market landscape, the increase in cybersecurity disclosures comes largely due to the guide of the Securities and Exchange Commission SEC [9]. It provides guidelines for the disclosure of cyberspace-related issues in 10-K presentations. Berkman et al. [10] and Ettredge et al. [11]. In 2018 SEC issued the second guidance on cybersecurity risk disclosure After three years, the US Securities Commission imposed the first sanctions for poor disclosure controls and procedures related to cybersecurity risks by issuing companies in June and August 2021, respectively [12,13].

In the European capital market, it can be said that there is mandatory disclosure of cyber incidents in the cybersecurity and privacy law, which includes the general data protection regulation (GDPR) and the directive on network and information systems security (NIS). According to Eijkelenboom and Nieuwesteeg [14], while this type of disclosure does not specifically require an incorporation in the annual financial report, the obligation to disclose data incidents from a cybersecurity law perspective could provide incentives for the board to also publish information on this topic in the report.

The previous literature on cybersecurity information disclosure has mostly been concentrated in the US stock market [10,11,15–19]. Studies in Canada were also presented [20,21]. In Europe, there was an investigation into cybersecurity disclosure in the Netherlands [14], and in the Asian continent, Barry et al. [22] studied the effect of institutional factors on information disclosures by Chinese companies.

Given that studies on the evaluation of cybersecurity disclosure in Latin America are non-existent, this research attempts to close this gap, taking the issuing companies of the main stock market indices of some countries in the region as object of study. According to the characteristics of our study, the content analysis technique focused on the examination and identification of information on cybersecurity that is disclosed voluntarily and mandatorily in annual reports and 20 F forms for Latin American companies during the period of 2016–2020. According to the previous literature related to cybersecurity disclosure from the content analysis methodology, only Héroux and Fortin [20] have proposed a disclosure index based on the guidelines of financial regulators of companies listed in the S&P/TSX 60 from the Toronto Stock Exchange (TSX). Other studies have used word counting techniques and textual analysis [3,10,14,17,18,21,23]. In this research, we have opted for the construction of a disclosure index, which is structured with 27 associated elements in four dimensions: governance, strategy, risk management, and financial implications. It was prepared based on previous literature and international standards related to corporate cybersecurity risk, and can thus be replicated in any specific context. The

purpose of this study was to create a cybersecurity disclosure index that allows analysis of the scope of the disclosure of voluntary and mandatory information on cybersecurity in Latin America. This article is organized as follows: first, advances in cybersecurity in both the international arena and Latin America are presented; then, the study methodology is exposed, which includes the presentation of the proposed disclosure index; and finally, the results of the application of the instrument are analyzed, and the conclusions of the investigation are presented.

## 2. Literature Review

### 2.1. Disclosure of Corporate Cybersecurity in the International Arena

The great challenges of cybersecurity have a global nature [24]. The intrusion of the digital realm into all areas of human activity, in conjunction with unprecedented levels of technological innovation and interdependence, have made it impossible for cybersecurity to be treated in isolation as a technical issue or separate policy area [25]. Cybersecurity efforts must be multidimensional in nature, due to the fact that a variety of factors are needed in order to build a resilient cybersociety [26], as well as a collaborative one, due to the fact that they require nations to work hand-in-hand with other countries [27].

Under this context, different cybersecurity measures have been taken worldwide, among which are the signing of the Budapest Convention in 2001, which marks the beginning of legislation on cybersecurity, as it is the first international treaty on violations committed in cyberspace. Subsequently, events such as the World Summits on the Information Society since 2003 have consolidated an international cooperation processes. In 2007, as an international framework, the World Agenda proposed ITU [28]: (i) the World Cybersecurity Index (IMC), among whose purposes is raising awareness of the role that governments play in promoting the culture of cybersecurity; (ii) the collection of information related to the CERT and CIRT regulations; (iii) national strategies, policies, certification, and awareness; and (iv) the ITU-IMPACT alliance, which generally focuses on scheduling cyber drills in different regions of the world in order to assess and improve response capacity in the case of cyber events.

In 2008, the ITU issued Recommendation ITU-T X.1205 as an international framework on general aspects of cybersecurity. Based on Resolution 64/25 of 2009, titled Advances in the field of information and telecommunications in the context of international security, the UN ratifies the invitation to its Member States, on the one hand, to the multilateral examination of threats in information security with new application strategies in the world, added to the evaluation of the inconveniences in information security; and on the other hand, to the mitigation and safeguarding of the circulation of information [29]. Later on in 2015, the Organization for Economic Cooperation and Development (OECD) issued a recommendation focused on managing digital security risks for economic and social prosperity [30]. At the same time, in order to comply with international provisions, different guidelines and regulations were issued in the United States and Europe—in most cases binding at the country level—measures that have been the object of convergence and adaptation at the global level.

An important reference for the disclosure of cybersecurity is the information to be reported required by the United States Securities and Exchange Commission (SEC). The SEC's first request for information obeys Article 503 (c) of Regulation SK [31], and requires disclosure and description of company-specific risk factors in the 10K section 1A report. This suggests that regulators assume that investors benefit from disclosures regarding company risks and uncertainties [32]. Six years later, in response to the growing impact of cyber incidents on clients and investors, the SEC issued the 2011 Cybersecurity Disclosure Guide; thus, the SEC visibly entered the cybersecurity arena, responding to concerns that public companies may not have provided adequate disclosures on cyber incidents [33]. This first guide provides the guidelines for disclosure of cyberspace-related issues on Form 10-K [10,34]. However, according to Amir et al. [15], if regulators want to ensure that

information about cyberattacks reaches investors, they should consider imposing stricter mandatory disclosure rules regarding cyberattacks and more clear materiality thresholds.

In February 2018, the SEC approved the second interpretive guidance on cybersecurity risk disclosure, emphasizing the importance of sufficient disclosure controls and procedures, as well as the prohibition of the use of non-public insider information from cyber incidents [21]. According to the SEC [9] and SEC [4] Corporate Finance Division, as with other operational and financial risks, registrants should continually review the adequacy of their disclosure in relation to the risks of cybersecurity and cyber incidents. The updated guidance adds two specific considerations [35]: (1) policies and controls put in place to detect and disclose significant cybersecurity risks and incidents; and (2) policies to prevent employees from dealing with non-public information regarding risks and cybersecurity incidents. According to Gao et al. [18], although the 2018 Guidance was unanimously approved by SEC commissioners, some felt that the new guidance did not go far enough. In addition to disclosing the risks of potential cyberattacks, companies should disclose known material cyber incidents that have already occurred, and discuss the potential costs and consequences [18]. According to Bakker [36], reporting on the consequences of a cybersecurity incident and the management of cybersecurity risks (including the identification and assessment of such risks and related liability) is of interest to stakeholders. The new guidance SEC [4] recommends that disclosures on cybersecurity risks and incidents in article 3D of Form 20 F be included in this section of the report. This measure is aimed at all foreign private issuers with capital shares registered in United States stock exchanges, and thus we can say that this is a requirement that has repercussions in the international context. In 2021, three years after the issuance of the Guide [4], the US Securities and Exchange Commission moved from orientation to application, increasing the requirements in cases where public companies do not implement robust cybersecurity risk management systems and related disclosure procedures [37]. Another possible North American reference to consider is the Canadian Securities Administrators [38], who, following in the SEC's footsteps, emphasize the importance of cyber risks and implementing cybersecurity measures to protect issuers and their stakeholders. The CSA provided issuers with references related to existing standards and publications, along with knowledge on the development of guides on cybersecurity and social media practices [20,39].

Regarding the European panorama, it can also be stated that although there is mandatory disclosure of cybersecurity incidents in accordance with the Cybersecurity and Privacy Law, the General Data Protection Regulation (RGPD), and the Directive on Network and Systems Security Information System (NIS), these guidelines do not specifically require incidents to be incorporated into the annual financial report; the obligations of disclosure of data incidents from the perspective of the Cybersecurity Law.

### 2.2. Cybersecurity in the Latin American Context

Although the governments of Latin America are aware of the need to protect the digital space upon which the functioning of society so greatly depends, cybersecurity has not gained a presence on the political agenda of the region with the urgency that would be expected [27]. Therefore, Latin American countries should continue to promote greater cooperation amongst themselves, involving all relevant stakeholders—as well as establishing mechanisms for monitoring, analysis, and impact evaluations related to cybersecurity—both at the national and regional levels [24].

Latin America has a high rate of connectivity, especially in urban areas where 71% of the population already has Internet access [40]. However, according to the Report on risks, progress and the way forward in Latin America and the Caribbean 2020, this strength contrasts with an incipient preparation to face the attacks that occur in cyberspace [24]. Aguilar [41] in reference to the Latin American and Caribbean region, states a lag in the construction of cyber-capabilities to face the current context, an environment that is full of risks from cyberspace and cyber threats. Raising awareness of cyber threats at the political level is only the first step in developing more harmonized cybersecurity capabilities in the

region; Latin American and Caribbean actors such as the Organization of American States (OAS) have already contributed to this process [24]. Thus, in 2004, the OAS Member States approved the comprehensive Inter-American Strategy to combat threats to cybersecurity with a multidimensional and multidisciplinary approach to the creation of a culture of cybersecurity and international cooperation for the region. Subsequently, through the Cybersecurity Program of the Inter-American Committee against terrorism of the OAS, technical capacity and the development of cybersecurity policies in the Americas are both strengthened.

With the impulse of the OAS, adherence to the European Union legislation on procedural matters on cybercrime has been achieved in Argentina, Chile, Colombia, Costa Rica, Panama, Paraguay, and the Dominican Republic [24]. Aguilar [42] highlights the importance of the region's national cybersecurity strategies being related to—or not distant from—the current understanding of cyber threats and the potential of cyberspace to violate public and national security. As of early 2020, twelve countries have approved national cybersecurity strategies, including Colombia (2011 and 2016), Panama (2013), Trinidad and Tobago (2013), Jamaica (2015), Paraguay (2017), Chile (2017), Costa Rica (2017), Mexico (2017), Guatemala (2018), the Dominican Republic (2018), Argentina (2019), and Brazil (2020), among others in progress [24]. According to OAS [43], it can be highlighted in this regard that Colombia was the first country in the region to develop a national cybersecurity strategy in 2011, in addition to being the first country to review it in 2016 with a comprehensive and collaborative approach, promoting management risk and multi-stakeholder engagement. The IDB OAS [44] confirm that seven of the thirty-two analyzed countries have a plan to protect their critical infrastructure, and that twenty of them have established some type of incident response group, called CERT or CSIRT, for its acronym in English. Twenty-two of these thirty-two countries are also considered to have little capacity to investigate crimes committed in cyberspace. Furthermore, that such crimes result in trials is still a greater challenge, for which it can be concluded that the region is not yet sufficiently prepared to face the attacks that occur in cyberspace [27].

In addition to commitments to developing cybersecurity capabilities at the national level, Latin America and the Caribbean have been the foundation for a series of highly dynamic regional initiatives [25]. For example, in 2016, CSIRT Americas was launched, a platform that allows regional cooperation and the exchange of information between the response teams to governmental and national incidents of the OAS Member States [25]. According to ITU and ABI Research [45], the Global Cybersecurity Index aims to provide a snapshot of where countries are in their cybersecurity commitments.

Figure 1 allows us to see the degree of compliance in cybersecurity based on the historical scores of the index for some Latin American countries for the years 2014, 2017, and 2018. During 2018, the role of Uruguay is highlighted by presenting the highest degree of compliance among Latin countries, such as Mexico in 2017 [46] and Brazil in 2014 [45]. Similarly, in the period of 2018, all nations that are an object of this study—according to the GCI—are categorized at a medium level, and more than half of countries have a low level of compliance [47]. In general, the commitment to cybersecurity in Latin America is below 0.70. The countries with the highest scores are Brazil and Uruguay. Colombia remained constant. Mexico improved to a high degree in 2017, and remained in 2018.

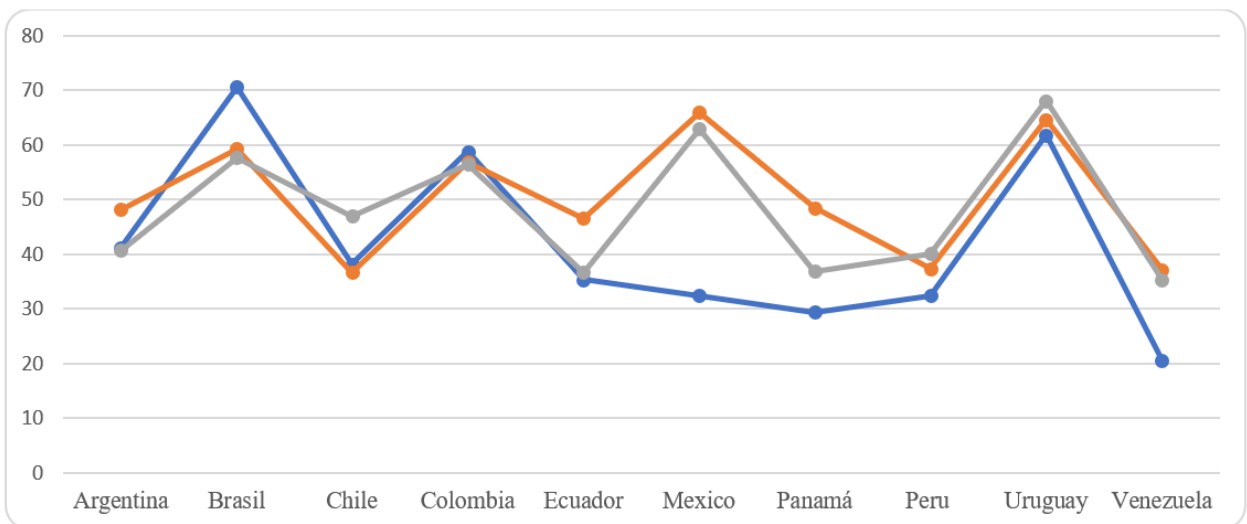

**Figure 1.** Latin American countries' Cybersecurity Index in 2014, 2017, and 2018. Note: these data were prepared based on [45–47].

As illustrated in Figure 2 below, the ITU report [48] measures the responsibility of Member States with cybersecurity. For its calculation, it maps eighty-two questions on the cybersecurity commitments of the Member States in five dimensions: (i) legal measures, (ii) technical measures, (iii) organizational measures, (iv) capacity development measures and (v) cooperative measures. Regarding the Global Cybersecurity Index for 2020, it can be seen that—in general—the highest level of development was found in Brazil (97.68), followed by Mexico (81.68), Uruguay (75.15), the Dominican Republic (75.07), Chile (68.83), Costa Rica (67.46), Colombia (63.74), Peru (55.68) and Argentina (50.12).

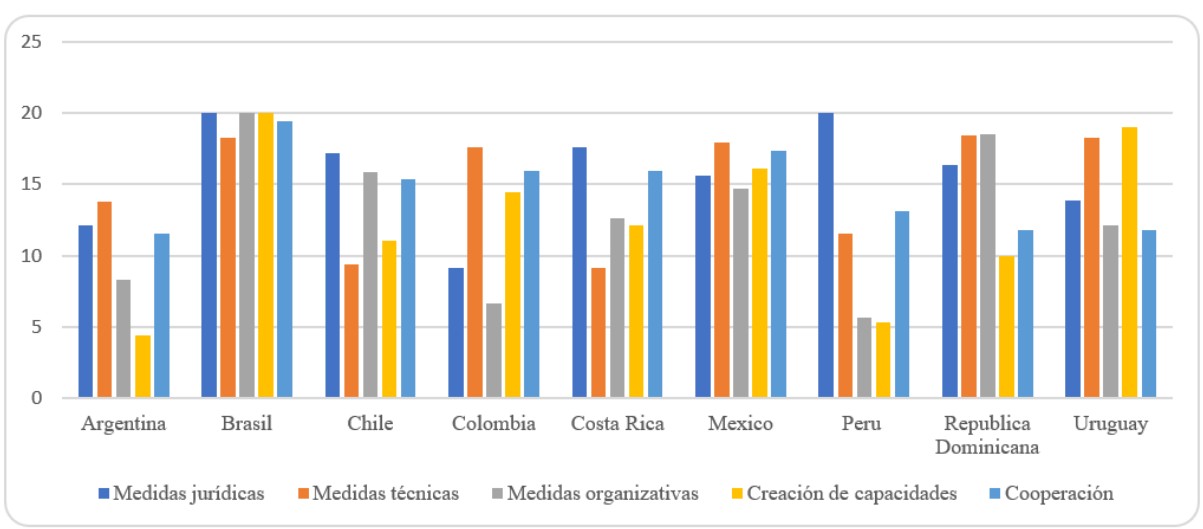

**Figure 2.** Latin American countries' Cybersecurity Index 2020 (by dimensions). Note: these data were prepared based on [48].

Reviewing Figure 2 highlights the legal measures issued in terms of cybersecurity, data protection, and critical infrastructures. At this point, the highest rates were in Brazil and Peru, with the highest score (20), and in Chile, with a high degree of compliance (17.2). Another pillar of importance is cooperation, relating to the evaluation of whether the countries have agreements (bilateral and multilateral) or alliances that contribute to the exchange of information, good practices, and uniformity in security measures; it is worth highlighting the scores of Brazil (19.41), Uruguay (19.04) and Mexico (17.34) here.

In the technical measures dimension, the Dominican Republic (18.42), Brazil (18.27) and Uruguay (18.27) once again had an important participation, standing out for their implementation and active status of the cybersecurity incident response teams (CIRT). Finally, regarding the degree of compliance with the capacity building pillar focused on training, awareness and capacity building initiatives in cybersecurity, the highest degree was presented by Brazil (20) and Mexico (16,13); and in the pillar of organizational measures focused on national cybersecurity agencies and strategies, the Dominican Republic (18.52) and Chile (15.84) rebounded.

### 2.3. Disclosure of Corporate Cybersecurity in Latin America

Our study was developed based on institutional theory considering that the countries of the region under study present their own factors that play a fundamental role in corporate practices, giving rise to the differences between countries. According to Barry et al. [22], the institutional environment for cybersecurity differs greatly from country to country. In this sense, advances related to cybersecurity in Latin America vary between countries, as illustrated in Figures 1 and 2. Thus, institutional factors condition corporate transparency, defined in this case as the availability of specific information from the listed company for those outside the company [49,50]. These institutional differences reflect a transfer of cybersecurity risk between governments and companies [49]. Scott [51] argued that the institutional framework leads to different organizational actions and strategies to obtain legitimacy. Along these lines, the disclosure of non-financial information of companies from different countries is affected by institutional factors, as corroborated by [22,52–55].

Although in the organizational sphere, Latin American companies do not refer to an exclusive regulation for the disclosure of cybersecurity information in the annual corporate report, the different regulations issued by the surveillance and control entities on corporate governance, risk management, and binding application of international standards on information security (such as ISO 27001) have become the subject of corporate disclosure in the region. Listed companies in Latin America that participate in the US stock market also report the risk factors related to cyber risk in the annual report 20 F item 3D, according to [4].

Another benchmark for Latin America that has generated an increase in the dissemination of cybersecurity is the adaptation of the European model of the regulations related to the protection of data privacy by issuing companies. We can also mention how the GRI standards promote cybersecurity disclosure—specifically with the GRI 418 Customer Privacy standard.

Based on the definition of critical infrastructure given by the United States Department of Justice (2001) [56], as the system and assets, whether physical or virtual, are so vital for each country that the incapacity or destruction of these systems and assets would have an impact undermining national economic security, public health and safety, or any combination of these, it is understood that some sectors are more vulnerable to cyber risk than others, including the communications, technology, energy, financial, health and food sectors, among others, presenting a latent threat. In the case of Latin America, according to the report IDB OAS [44], the financial sector is a frequent target. This is how the digitization of this industry must consider the management of cybercrimes, since establishing digital environments also implies analyzing cyber risks that must be avoided and managed properly [43]. As corroborated by Porrúa and Contreras [24], cyberattacks in the region have been on the rise, and mainly targeting financial institutions in Latin America. Previous studies, such as Freedman and Jaggi [52] and Prado-Lorenzo et al. [53], showed that companies from various sectors present important differences in the scope of disclosure of non-financial information.

In general terms, disclosure on cybersecurity is voluntary in Latin American countries; however, the disclosures regulated by the SEC in Form 20F number 3 D are mandatory for listed companies, as follows (Table 1):

**Table 1.** Voluntary and mandatory information disclosed by listed companies in Latin America.

| Voluntary Disclosure of Cybersecurity Information | Mandatory Disclosure of Cybersecurity Information |
| --- | --- |
| Voluntary disclosure on cybersecurity is related to the regulations and advances that have been presented on cyber risk management at the national level based on international standards of global application with effects on the business world: the national cybersecurity strategy, the protection of data law, good corporate governance codes, and financial sector regulations related to cybersecurity. GRI standards also promote disclosure related to cybersecurity, in a timely manner. GRI 418 Customer privacy. | Mandatory cybersecurity disclosure relates to risk factors. Item 503(c) of Regulation S-K and Item 3D of Form 20-F require companies to disclose the most significant factors that make investments in the company's securities speculative or risky. Companies should disclose the risks associated with cybersecurity and cybersecurity incidents if these risks are among such factors, including risks that arise in connection with acquisitions. |

*2.4. Previous Studies on Cybersecurity Disclosure*

While cybersecurity risks are significant and could materially affect business operations and the integrity of financial reporting, there is limited empirical research on cybersecurity risk disclosure trends and practices of listed companies [18]. In addition to disclosing the risks of potential cyberattacks, companies should disclose known material cyber incidents that have already occurred and discuss the potential costs and consequences [4].

Some studies related to cybersecurity disclosures have used textual language analysis based on corporate disclosures on Form 10K: reference [17] developed an index that captures the existence of voluntary disclosures related to information security and exhibits a positive association between the market valuation and its measurement; reference [10] developed an index that captures the existence of voluntary disclosures related to information security and exhibits a positive association between the market valuation and its measurement; reference [19] demonstrated that there are different perceptions among investors regarding the impact of security incidents depending on the nature of the disclosure and presented outlines of how market participants can better interpret the disclosed security risk factors; reference [54] examined the utility of cybersecurity-related risk factors disclosed in the 10K submissions, documenting that the presence of these factors in the period prior to SEC guidance is related to future reported cybersecurity incidents; their findings generally support the SEC's decision to emphasize cybersecurity risk disclosure.

According to reference [20], risk factor disclosure accounts for 7% of the average length of a 10K presentation per page count. In accordance with [18], the two cybersecurity risks most frequently revealed by listed companies are those related to the interruption of service and the loss of confidential data. In addition, Article 1A of the 10K Report is the most widely used place of disclosure; however, some companies also use Articles 1 and 7 to disclose regulatory risks and cyber incidents, respectively. Another investigation into cybersecurity disclosures [33] assessed the conditions under which US federal government incentives/regulations, such as the Sarbanes–Oxley Act of 2002 and the SEC's Disclosure Guidance on Cybersecurity Risks and Incidents 2011, led to increased investments in cybersecurity and, in turn, to a greater disclosure of said information [16]. Based on a qualitative analysis of five corporate cases of data breaches in the US from a systematic post-disclosure review of cyber risk, they determined a very limited reaction in share prices associated with the disclosure of repetitive and nonspecific cyber risks. Reference [15], using voluntarily disclosed data on cyberattacks and those that were withheld and later discovered by external sources, estimated the extent to which companies retain information on cyberattacks and its association with a decline in the value of shares in the month in which the attack is discovered.

Recently, reference [20] examined whether the content of cybersecurity disclosures of Canadian companies listed in the S&P/TSX 60 index is aligned with best practices. Based on a self-constructed index with 40 items, they concluded that cybersecurity disclosure levels are low and suggested that financial regulators could issue stricter requirements.

Reference [14] provided an exploratory empirical analysis of cybersecurity information disclosure, through the annual reports of listed companies in the Netherlands in 2018. Lastly, reference [21] assessed how the gender composition of the board may influence the scope of such disclosure, based on a sample of companies included in the S&P/TSX 60 index during the 2014–2018 period; their results showed evidence of a positive association between the presence and level of cybersecurity disclosure and the gender diversity of the board.

## 3. Methodology

### 3.1. Study Population and Sample

As the population under study, the listed companies in the main stock market indices of the Latin American countries with the largest capitalization in the region were selected: Argentina, Brazil, Chile, Colombia, Mexico and Peru, which capture more than 70% of the market in the region [57].

Four economic sectors were studied: energy, financial, consumer discretionary and materials, under two criteria: (i) their high vulnerability to cyber risk, and (ii) the presence of companies from each sector in most of the countries in the sample. For grouping purposes, companies were analyzed by the country and sector of economic activity according to the classification proposed by the Global Industry Classifications Standards (GICS). The final sample yielded a total of 425 observations in the period of 2016–2020 (Table 2).

**Table 2.** Sample of companies by country and stock index.

| Country | Index | N° Companies |
|---------|-------|--------------|
| Argentina | S&P Merval | 7 |
| Brazil | Ibovespa | 25 |
| Chile | IPSA | 18 |
| Colombia | Colcap | 16 |
| Mexico | S&P/BMV IPC | 11 |
| Peru | S&P/BVL Peru | 8 |
| Total | | 85 |

The data were collected in the first half of 2021, based on the content analysis of the annual reports published on the companies' websites and the 20 F report, in the case of the companies in the sample issuing in the United States regulated by SEC. The measurement of the scope of the disclosures was carried out taking the proposed disclosure index as a reference; it was initially performed by one of the authors, and was subsequently independently reviewed by another. The companies that did not provide information on cybersecurity have not been considered in the study, a situation that has significantly decreased the sample.

### 3.2. The Proposed Cybersecurity Disclosure Index

Although previous studies related to cybersecurity disclosure have mostly utilized word counting and textual analysis, in this investigation, we have opted for the construction of a disclosure index. We agree with Marston and Shrives [58] that measuring the disclosure of information by counting data elements is not a satisfactory solution to the research problem, due to the fact that there are repetitions of certain data, numbers, and words in the annual reports and other means of disclosure of organizational information. The literature review corroborates how content analysis is the prevailing methodology, regardless of the technique used to collect the information. According to Berelson [59], it aims to be objective, systematic, and quantitative in the study of the manifest content of communications. Its use in the evaluation of cybersecurity disclosures is presented in [10,14,17,18,20,21,60–62]. These investigations mostly evaluate the information reported on cybersecurity by listed companies in the reports filed with the SEC.

A disclosure index is one of the main methods of evaluating the information transparency of public and private institutions [63]. Marston and Shrives [58] state that, from 1960 to the present, the disclosure index has been a research tool that persists over time and represents extensive lists of selected items that can be revealed in company reports. Returning to Ortiz and Clavel (2006) [64], the construction of disclosure indices is a highly widespread practice; its elaboration is part of one aspect of the content analysis methodology, and is one of the main techniques used to study the information provided by the public and private organizations. Previous studies related to issues of disclosure of non-financial information confirm its wide acceptance in research [53,65–80].

Table 3 below presents two previous studies that propose a cybersecurity disclosure index: one prepared based on the guidelines of financial regulators of companies listed on the S&P/TSX 60 index of the Toronto Stock Exchange (TSX), and another prepared by the international accounting and auditing firm E&Y [8,81,82], which seeks to analyze the disclosures related to cybersecurity in the declarations and presentations of the Form 10-K of the Fortune 100 companies to identify trends and emerging developments.

**Table 3.** Instruments for measuring cybersecurity disclosure according to previous literature.

| Authors | Country | Dimensions |
|---------|---------|------------|
| Héroux and Fortin [20] | Canada | The index is made up of forty elements, distributed in seven categories: (i) risk factors (4 items); (ii) potential impacts (11 items); (iii) responsibility for the cybersecurity strategy (6 items); (iv) risk mitigation (11 items); (v) possible cybersecurity incidents (2 elements); (vi) actual cybersecurity incidents (3 items); and (vii) other disclosed cybersecurity elements (3 elements). Prepared based on CSA, 2017; SEC, 2018b. |
| E&Y [8,81,82] | EE.UU. Fortune List 100 | The measuring instrument consists of several dimensions: Board oversight (Risk oversight (1), Board level (3), Reporting management (2), Reporting frequency management (2)); Cybersecurity risk statements; Risk management (cybersecurity risk management efforts (3)); Education and training (1). |

In the present study, we have opted for an unweighted index in order to avoid the arbitrariness that is inherent in the use of any weighted index referred to by Giner [83], and it assumes that each disclosed element has the same importance [84]. According to Choi [85] or Chow and Wong-Boren [86], studies that use weighted and unweighted indices obtain similar results. The proposed index aims to assess the disclosure of information on cybersecurity. It is structured in four dimensions, based on the following references: (i) governance [8,38,87]; (ii) strategy [20,87]; (iii) risk management [8]; and (iv) financial implications [4]. The cybersecurity disclosure measure is based on 27 individual elements, each of which refers to a particular topic associated with one of the four dimensions, as follows: governance (5 elements), strategy (6), risk management (13), and financial implications (3).

The selection of the items that make up the Cybersecurity Disclosure Index (CDI) was made from [8,20,87] and some previous research on the international standards related to the management of the corporate information system: the ISO 27,000 family of standards, the guidelines of the SEC [4,9], the General Data Protection Regulation (GDPR) of the European Union, and other documents proposed by entities with broad international recognition, such as the Organization for Economic Cooperation and Development (OECD), the Inter-American Development Bank (IDB), the Organization of American States (OAS), and the Initiative of Global Report (GRI), among others.

To validate the consistency of this index, Cronbach's Alpha coefficient has been calculated using the correlation matrix to check the reliability of the instrument items. According to Table 4, this coefficient yields a result of 0.8168, thus confirming that the CDI is reliable and consistent with the items selected from the sample.

**Table 4.** Cronbach's Alpha.

| Dimension | Obs | Sign | Item–Test Correlation | Item–Rest Correlation | Average Interitem Correlation | Alpha |
|---|---|---|---|---|---|---|
| Governance | 425 | + | 0.8104 | 0.6485 | 0.5197 | 0.7645 |
| Strategy | 425 | + | 0.8182 | 0.6614 | 0.5113 | 0.7584 |
| Risk | 425 | + | 0.862 | 0.7358 | 0.4644 | 0.7223 |
| Financial | 425 | + | 0.7229 | 0.5119 | 0.6133 | 0.8263 |
| Test scale | | | | | 0.5272 | 0.8168 |

The CDI unweighted disclosure index for each company is expressed as follows:

$$CDI = \frac{\sum_{i=1}^{t} c_i}{t} \tag{1}$$

where:

$c_i = 0$ or 1 according to the following conditions:

$c_i = 0$ if the disclosure item was not found;

$c_i = 1$ if the disclosure item was found.

$t =$ the maximum number of cybersecurity disclosure items that a company could disclose.

The following is the composition of the Cybersecurity Disclosure Index (CDI) for each dimension (Table 5):

**Table 5.** Cybersecurity disclosure index.

**Governance [8,39,87]**

G1. Describes the involvement of the board of directors in overseeing cybersecurity-related risks and opportunities [4,8,87].
G2. Reveals the existence of a board-level committee specifically charged with cybersecurity and/or information security [8].
G3. Discloses the existence of a committee in charge of supervising cybersecurity differently from the audit committee, e.g. technology, risks [8,20].
G4. Discloses oversight of cybersecurity matters by an audit committee [8,20,88].
G5. Reports on the contribution of management to the design and evaluation of the organization's information security risk management system [87].

**Strategy [20,87]**

E1. Disseminates the formulation of a cybersecurity policy(s) aimed at managing information security [89].
E2. Discloses the existence of a security management system in the organization [90].
E3. Reveals that the information security management system is established in accordance with internationally-recognized norms and standards (national regulations require the application of these standards in some sectors).
E4. Reveals the existence of a personal data protection policy and/or guarantee of digital rights [20].
E5. Discloses the application of cybersecurity awareness and training strategies for members of the organization to mitigate cybersecurity risk [8,20].
E6. Discloses the existence of communication procedures that provide information on cybersecurity to decision makers, customers, employees, other market participants and regulators, as appropriate [91].

**Risk management [8]**

R1. Presents a description of cybersecurity and/or information security risks [20,90].
R2. Includes data protection as a material topic [92].
R3. Includes cybersecurity and/or information security as a material topic [92].
R4. Includes cybersecurity and/or information security as a risk factor [4,8,93]
R5. Includes data privacy as a risk factor [4,8].
R6. Reveals cybersecurity risk within other organizational risks [93].
R7. Reports on response procedures to incidents in the information systems (contingency plan) [39].
R8. Reveals the development of tests and monitoring to validate the effectiveness of cybersecurity policies and procedures [9,10,93]
R9. Presents prior information on ongoing cybersecurity incidents or other past events that may be relevant as a risk factor for the organization [4].
R10. Discloses the frequency of management reports to the board or committee [8].
R11. Discloses claims regarding privacy violations and loss of customer data [91,94,95]
R12. Reports on the participation of the internal audit in the management of cyber risk and/or information security [8].
R13. Reports on the participation of an independent external advisor in cybersecurity risk mitigation [8].

**Financial implications [4]**

I1. Mentions insurance coverage related to cybersecurity or payments to service providers [4].
I2. Discloses information related to the measurement of ongoing cybersecurity efforts [4].
I3. Reports on the measurement of the costs, consequences, and risks of cybersecurity incidents [4].

The dimensions that make up the proposed cybersecurity index are presented below:

Dimension 1. Governance:

Cybersecurity governance, as part of information security governance, is the process of directing and controlling the protection of a company's digital information assets against risks related to the use of the Internet [96]. ISO [97] defines this as the system by which the information security activities of an organization are directed and controlled. Therefore, when it comes to cybersecurity governance, the most important thing that a board can do is set the right tone and align the appropriate risk appetite related to cybersecurity with management. Risk assessment and the development of mitigation principles to manage them are likely to be effective, so long as there is a well-communicated and coordinated governance policy within the system being managed [7]. In its approach, the board of directors, management, business unit leaders, and security and IT groups should all participate [98]. Everyone involved in the daily operation of the system in question must understand that in order for security to be effective, it must be part of the daily operating culture [7]. In this context, stakeholders want to better understand how companies are preparing for—and responding to—cybersecurity incidents, as well as how boards of directors oversee these critical risk management efforts [81].

Against this background, the Cybersecurity Disclosure Guide [4] sets out how disclosure about the board's involvement in oversight of the risk management process should provide important information for investors regarding how a company perceives the role of the board and the relationship between the board of directors and senior management in the management of the material threats it faces. According to [81], it is also in the interest of investors to understand whether the full board is in charge of supervision, shares it between the different audit, risk, and technology committees, or creates a committee that is exclusively dedicated to cybersecurity. According to Ernst and Young [99], the board's oversight functions include requesting management to review cybersecurity disclosures over the past two to three years with benchmarking.

Dimension 2. Strategy:

A personalized, business-driven, and threat-based cyber strategy enables organizations to focus on security weaknesses and reduce the likelihood of future financial or reputational damage [100]. Considering that two of the board's main responsibilities are strategy and cybersecurity risk management [82], the proposed disclosure index defines strategy and risk management as two dimensions, themes that represent the elements of how organizations operate [87]. In this section, the business-oriented cybersecurity strategy dimension is presented as the most important step for security leaders amid accelerated business digitization [101].

From this perspective, the disclosure guide SEC [4] highlights the importance of maintaining comprehensive policies and procedures related to cybersecurity risks and incidents, and AICPA [90] ratifies the option to disclose a narrative description that includes these policies in corporate reports. Key security processes are implemented and operated in order to protect information and systems against those risks. Among the policies disclosed by companies is that related to the protection of personal data and/or guarantee of digital rights and the cybersecurity and/or information security policy.

According to AICPA [90], management efforts have led to the development of numerous risk management frameworks that aim to provide guidance to organizations on how to manage cybersecurity risk (for example, ISO/IEC 27,001 [89] and the cybersecurity framework of NIST). From this perspective, companies try to design and implement effective programs, a situation that deserves to be communicated to interested parties, along with the controls and processes that protect their operations, such as training and evaluation [99] to members of the organization and communication procedures that provide information on cybersecurity to decisionmakers, customers, employees, other market participants and regulators, as appropriate.

Dimension 3. Risk management:

Information security risk management is the general process that integrates the identification and analysis of the risks to which the organization is exposed, as well as the evaluation of the potential impacts on the business and the decision of what actions can be taken to eliminate or reduce the risk to an acceptable level [102]. Disclosure policies and procedures should ensure that information on cybersecurity risks and incidents is processed and disclosed [98]. Understanding the information conveyed by cybersecurity risk disclosures is important, as it can help investors assess a company's cybersecurity risk and provide regulators with information on whether additional legislative standards are necessary for encouraging companies to disclose more about their cybersecurity risks [54]. The purpose of providing risk factor disclosures is to discuss the most important elements that make the company risky [103]. Companies should disclose known material cyber incidents that have already occurred, and discuss the potential costs and consequences [18].

Regarding the protection of customer privacy, this is an objective that is generally recognized in national legislation and organization policies [94]. As stipulated in the OECD Guidelines for Multinational Enterprises, organizations are expected to respect consumer privacy and take reasonable steps to ensure the security of the personal data they collect, store, process or disseminate. The internal audit function can play an important role, both in providing assurance regarding information security and in generating insights on how to improve the organization's information security [104]. It may also be helpful to seek input from external specialists in cybersecurity assessment. Companies can conduct annual external reviews of security and privacy programs, including incident response, breach notification, disaster recovery, and crisis communication plans [105]. Third-party security assessments can also provide benchmarking relative to other companies of similar size, or that exist in the same industry [105].

Dimension 4. Financial implications:

It is essential that boards of directors understand, from risk management, the probability that the risk will occur and the estimate of the financial cost of the damages that would result [82]. Cybersecurity—as well as physical threats to the network infrastructure—can have significant economic repercussions, and therefore, investing in data security and upgrading the network infrastructure is crucial [4,92]. Companies are expected to disclose cybersecurity risks and incidents that are important to investors, including the attendant financial, legal, or reputational consequences. Cybersecurity insurance helps companies transfer a portion of the potential risk and exposure associated with cybersecurity incidents [21]. Companies are expected to disclose cybersecurity risks and incidents that are important to investors, including entailed financial, legal consequences or reputation risks.

## 4. Results

During the period under study, a sustained growth was observed in the average cybersecurity disclosure index for all companies in the sample (Table 6). However, the annual average score does not exceed 40% of the total score, a situation that denotes the possibility of significantly expanding the dissemination of information on cybersecurity in Latin American companies in future periods.

If we analyze the detail of the indices by country, we observe that Argentina is presented as the nation that reports the highest disclosure index during the five years under study. The score one obtains can be due to different factors. One of these factors is that 57% of the companies in the sample belong to the financial sector and, when comparing the score obtained by the companies in this sector with that of the other sectors, a significant difference is obtained in this country. We can also observe that 86% of organizations in Argentina report information to the SEC on Form 20 F. It should additionally be noted that at the national level, Argentina has a cybersecurity strategy and data protection policy.

**Table 6.** Cybersecurity information disclosure by country and sector.

| Year | 2016 | 2017 | 2018 | 2019 | 2020 |
|---|---|---|---|---|---|
| | | | Country | | |
| Argentina | 0.30 | 0.38 | 0.41 | 0.47 | 0.51 |
| Brazil | 0.17 | 0.23 | 0.27 | 0.35 | 0.40 |
| Chile | 0.13 | 0.18 | 0.25 | 0.35 | 0.43 |
| Colombia | 0.18 | 0.23 | 0.31 | 0.33 | 0.38 |
| Mexico | 0.10 | 0.17 | 0.25 | 0.28 | 0.40 |
| Peru | 0.17 | 0.22 | 0.22 | 0.24 | 0.25 |
| | | | Sector | | |
| Discretionary consumer | 0.11 | 0.17 | 0.21 | 0.26 | 0.28 |
| Energy | 0.15 | 0.18 | 0.21 | 0.30 | 0.35 |
| Financial | 0.28 | 0.34 | 0.42 | 0.45 | 0.52 |
| Materials | 0.10 | 0.12 | 0.16 | 0.20 | 0.28 |

In general, it can be seen that Latin American countries present important advances on the issue of national cybersecurity. However, the greatest increase in disclosures in the period of analysis is presented by Chile and Mexico, reaching an increase of 0.30 between 2016 and 2020. In the case of Argentina, it went from 0.30 in the year 2016 to 0.51 in 2020, corresponding to an increase of 0.21. For Brazil, the range of the cybersecurity disclosure index went from 0.17 in 2016 to 0.40 in 2020, reaching an increase of 0.23. Likewise, Colombia went from 0.18 in 2016 to 0.38 in 2020. Finally, there was a lower increase in Peru, at 0.08, since its cybersecurity disclosure index was registered at 0.17 in 2016, and 0.25 in 2020.

Interpreting these results requires the consideration of two events that have marked the future of cybersecurity in the countries under study in the period of 2016–2020. First, the approval of the National Cybersecurity Strategy: in Colombia (2016 version), Chile (2017), Mexico (2017), Argentina (2019) and Brazil (2020). Peru does not yet have a national strategy. Second, the issuance of data protection laws: in Argentina (2016), Brazil (2018), Colombia (2018), Chile (1999), Mexico (2017) and Peru (2011). Another event that generated important changes in regional cybersecurity was the WannaCry ransomware cyberattack in 2017, from which—according to Creese [26]—CSIRT Americas facilitated the early identification and isolation of hotspots of infection in Latin America to stop the spread of WannaCry within the region. Thus, the platform has created a central repository of tools for its regional components to prevent and combat ransomware infections by mitigating future outbreaks. Finally, the crisis caused by the COVID-19 pandemic during 2020—in addition to exposing the structural deficiencies that our society has been carrying in multiple systems—has also highlighted the catalytic role of technology in the way that we have collectively faced the pandemic Barmpaliou [25]. In this way, each and every one of the measures that have been adopted to mitigate cyber risk during this crisis have been converted into actions that are in favor of both present and future cybersecurity.

Analyzing the details of the indices by sector, we observe that the companies belonging to the financial sector are those that reported, on average, the highest disclosure index during the five years under study. In addition, there was a high growth in cybernetic information disclosed by companies that have carried out this type of activity since 2016 (0.28) until 2020 (0.52). As IDB OAS [44] states, the financial sector is more advanced in cybersecurity because it is a frequent target, and therefore, it is investing more in cybersecurity. This is corroborated by Porrúa and Contreras [24]: cyberattacks in the region have been increasing, mainly targeting financial institutions in Latin America.

Regarding the energy sector, a score was obtained that oscillated in a range of 0.15 in 2016 and 0.35 in 2020. In third place is the consumer discretionary sector, with a score of 0.11 for 2016 and 0.28 for 2020; and finally, the materials sector reported the lowest scores on cybersecurity disclosure.

Regarding the longitudinal analysis of the dimensions of the cybersecurity information disclosure index (Table 7), a sustained growth is observed in the score obtained in each of the dimensions, which was especially important in the years 2019 and 2020. Regarding the dimension of the index that shows greater disclosure on the subject of study in 2020, the strategy stands out (0.53), followed by all those activities that serve and contribute to cybersecurity risk management (0.40). Third is governance (0.36), and finally, disclosure on financial implications (0.21). Regarding this last dimension, the increase in its score began in 2018, which can be related to the implementation of the guidelines of the Disclosure Guide SEC [4], as well as to the measures implemented due to the cyberattack WannaCry in 2017.

**Table 7.** Cybersecurity disclosure index by dimension.

| Dimension/Year | 2016 | 2017 | 2018 | 2019 | 2020 |
|---|---|---|---|---|---|
| Governance | 0.13 | 0.19 | 0.24 | 0.29 | 0.36 |
| Strategy | 0.24 | 0.31 | 0.39 | 0.47 | 0.53 |
| Risk management | 0.18 | 0.23 | 0.28 | 0.34 | 0.40 |
| Financial implications | 0.03 | 0.07 | 0.12 | 0.15 | 0.21 |

The analysis of the data contained in Table 8—in which the score of the scope of cybersecurity disclosure is presented in the 27 items observed—allows the following results to be obtained, depending on the observed dimension.

**Table 8.** Cybersecurity disclosure by dimension.

| | 2016 | 2017 | 2018 | 2019 | 2020 |
|---|---|---|---|---|---|
| | | | Governance | | |
| G1 | 0.18 | 0.24 | 0.35 | 0.42 | 0.53 |
| G2 | 0.11 | 0.16 | 0.16 | 0.19 | 0.24 |
| G3 | 0.18 | 0.24 | 0.26 | 0.32 | 0.38 |
| G4 | 0.02 | 0.06 | 0.09 | 0.13 | 0.20 |
| G5 | 0.19 | 0.25 | 0.34 | 0.41 | 0.45 |
| | | | Strategy | | |
| E1 | 0.24 | 0.36 | 0.44 | 0.52 | 0.58 |
| E2 | 0.46 | 0.53 | 0.64 | 0.68 | 0.68 |
| E3 | 0.13 | 0.20 | 0.26 | 0.33 | 0.39 |
| E4 | 0.19 | 0.24 | 0.34 | 0.42 | 0.48 |
| E5 | 0.29 | 0.38 | 0.46 | 0.58 | 0.72 |
| E6 | 0.11 | 0.13 | 0.21 | 0.27 | 0.33 |
| | | | Risk management | | |
| R1 | 0.32 | 0.40 | 0.46 | 0.52 | 0.59 |
| R2 | 0.14 | 0.19 | 0.25 | 0.34 | 0.44 |
| R3 | 0.09 | 0.16 | 0.26 | 0.33 | 0.48 |
| R4 | 0.40 | 0.44 | 0.49 | 0.56 | 0.58 |
| R5 | 0.25 | 0.31 | 0.33 | 0.41 | 0.44 |
| R6 | 0.18 | 0.25 | 0.33 | 0.39 | 0.44 |
| R7 | 0.15 | 0.18 | 0.19 | 0.24 | 0.36 |
| R8 | 0.22 | 0.32 | 0.35 | 0.41 | 0.47 |
| R9 | 0.06 | 0.15 | 0.19 | 0.18 | 0.24 |
| R10 | 0.02 | 0.06 | 0.07 | 0.11 | 0.14 |
| R11 | 0.31 | 0.35 | 0.38 | 0.42 | 0.45 |
| R12 | 0.12 | 0.14 | 0.21 | 0.29 | 0.34 |
| R13 | 0.04 | 0.09 | 0.14 | 0.16 | 0.26 |
| | | | Financial implications | | |
| I1 | 0.06 | 0.12 | 0.18 | 0.20 | 0.28 |
| I2 | 0.02 | 0.04 | 0.12 | 0.19 | 0.21 |
| I3 | 0.01 | 0.04 | 0.06 | 0.05 | 0.15 |

Dimension 1. Governance

Regarding the disclosure of the participation of the board of directors in the supervision of risks related to cybersecurity (G1), the score increases from 0.18 for 2016 to 0.53 for 2020. These scores reflect the important participation in which the board is assuming directive on security risk management and management shows willingness to disclose it. Regarding the disclosure of the existence of a board-level committee specifically in charge of cybersecurity and/or information security (G2), it can be seen that its score was 0.11 in 2016 and 0.24 in 2020, meaning that they have carried out actions aimed at working on cybersecurity from the corporate government by creating the committees that will exclusively manage cyber risk. On the disclosure of the existence of a committee in charge of supervising cybersecurity other than the audit committee, G3 yielded values of 0.18 in 2016 and 0.38 in 2020. The next item, G4, disclosed the supervision of cybersecurity matters by an audit committee—0.02 in 2016 and 0.20 in 2020. These results show that the audit committee has, in some cases, been leading the issue of cybersecurity. The last item of the governance dimension, which considers whether the contribution of management is reported in the design and evaluation of the organization's information security risk management system (G5), presented a score of 0.19 in 2016 and 0.45 in 2020. This latest evaluation shows how management's participation has been the subject of greater disclosure in recent years.

Dimension 2. Strategy

Regarding the dissemination of the formulation of a cybersecurity policy(s) aimed at managing information security (E1) in Latin America, 2016 presented a score of 0.24, and 2020 presented a score of 0.58, being able to show how it has been sought out over time to reveal corporate cybersecurity policies.

The evaluation of the following item discloses the existence of an SGS security management system in the organization (E2), and presented a score of 0.46 in 2016 and 0.68 in 2020, which may mean that the organizations are interested in communicating the implementation and management of SMS as a strategy for mitigating cyber risk. At the same time, we also sought to evaluate the disclosure on whether the management system has been developed under internationally-recognized norms and standards (E3). This item reported a score of 0.13 in 2016 and 0.39 in 2020.

Faced with the disclosure of the existence of a personal data protection policy and/or guarantee of digital rights (E4), it can be seen that this item scored 0.19 in 2016 and 0.48 in 2020. As already stated regarding some countries in the period of 2016–2020, laws related to data protection have been created, a factor that can promote disclosures on the subject. Faced with the disclosure on the application of cybersecurity awareness and training strategies of the members of the organization to mitigate cybersecurity risk (E5) in 2016, a score of 0.29 was obtained, as well as a disclosure score of 0.72 in 2020. Item E6 also reveals the existence of communication procedures that provide information on cybersecurity to decisionmakers, customers, employees, other market participants and regulators, as appropriate, and obtained a score of 0.11 in 2016 and 0.33 in 2020.

Dimension 3. Risk management

Regarding cybersecurity risk management, 13 items were evaluated. The first of them (R1) offers a clear increasing provision to describe cybersecurity and/or information security risks, from 0.32 in 2016 to 0.59 in 2020. Although there was an increase in the majority of observations, the inclusion of data protection as a material issue (R2) did not reach the average in 2020, 0.44, in the same way that cybersecurity and/or information security as a subject material (R3) presented an increasing trend from 0.09 in 2016 to 0.48 in 2020; and the risk factor, R4, obtained a disclosure percentage in 2020 of 0.58. For its part, data privacy as a risk factor in organizations (R5) also showed an increase that reached 0.44 in 2020.

In this same orientation, the disclosure of information from the studied sample revealed a growth in focus on cybersecurity in the risk supervision section (R6) from 0.18 in 2016 to 0.44 in 2020. In the disclosure of information on response procedures to incidents in information systems—that is, the existence of a contingency plan (R7)—there was an

increase of 0.21 under a growing trend from 0.15 in 2016 to 0.36 in 2020. On the other hand, regarding the methods of avoiding vulnerabilities, there was a growing development, from 0.22 in 2016 to 0.47 in 2020, in the execution of tests and monitoring to validate the effectiveness of the cybersecurity policies and procedures (R8) that organizations implemented.

In general terms, the item disclosure of prior information on ongoing cybersecurity incidents or other past events that may be relevant as a risk factor for the organization (R9) in 2016 obtained 0.06, and its maximum disclosure only reached 0.24 in 2020. Regarding the disclosure of the item examining the frequency of management reports to the board or committee (R10), the variation ranged between 0.02 for 2016 and 0.14 for 2020. Claims related to violations of privacy and loss of customer data (R11) (GRI [94]; OECD, [95]) have increased from 0.31 in 2016 to 0.45 in 2020, with a significant degree of disclosure. Regarding the participation of the internal audit in the management of cyber risk and/or information security (R12), 0.12 was disclosed in 2016, and 0.34 in 2020. Finally, the participation of an independent external advisor in cybersecurity risk mitigation (R13) began with a score of 0.04 in 2016 and ended with a score of 0.26 by 2020.

Dimension 4. Financial implications

Regarding the dimension of financial implications, this is the dimension with the least disclosure in the period under study. Regarding the first element, which has to do with insurance coverage related to cyber security or payments to service providers (I1), in 2016, it was reported at 0.06, and for 2020, at 0.28, with this being the most significant component of the three analyzed. Regarding the second item—information related to the measurement of continuous efforts in cybersecurity (I2)—the increase during the five years has been 0.19, going from 0.02 in 2016 to 0.21 in 2020. Finally, the item with the least disclosure in the financial implications dimension was the measurement of the costs, consequences and risks of cybersecurity incidents (I3), only obtaining a score of 0.01 in 2016 and 0.15 in 2020.

The analysis of the data contained in Table 8 shows the effort of Latin American companies to reveal the implementation of corporate cybersecurity measures. It can be seen that the most publicized item in 2020 is the one related to the application of cybersecurity awareness and training strategies to the members of the organization to mitigate cybersecurity risk (E5), which, with a score of 0.72, experienced a variation of 0.43 since 2016. The second element that registered the greatest increase in the level of disclosure is the dissemination of the formulation of a cybersecurity policy(s) oriented toward the management of information security (E1), with an increase of 0.34 points from 2016 to 2020. The third item with a significant increase in disclosure was the existence of a personal data protection policy and/or guarantee of digital rights (E4), with an increase of 0.29 between 2016 and 2020. Regarding the disclosure of the year 2020, two important values are observed, one of them from item E5, which discloses the application of awareness and training strategies in cybersecurity for members of the organization to mitigate cybersecurity risk, and item E2, which reveals the existence of a security management system in the organization, with scores of 0.72 and 0.68, respectively. It is noteworthy that most of the items that have experienced a greater increase in their score, and that obtained a greater percentage in 2020, belong to the dimensions of strategy and risk management, showing that companies give greater relevance to the disclosures that take part in cybersecurity actions designed over the long term whose objective corresponds to the mitigation of cyber risk.

## 5. Discussion and Conclusions

In Latin America, actions on cybersecurity are promoted by the OAS with different initiatives that promote cybersecurity and international cooperation for the region. At the national level, countries have gradually adhered to international policy measures. Some have made important progress, such as the ratification of the Budapest Convention, the approval of the national strategy, and the creation of laws on data protection, among others. This is the reason why the measurement of national cybersecurity commitment in 2020 shows a better score, compared to that of previous years in the different dimensions that make up the Global Cybersecurity Index—legal, technical, organizational measures,

capacity building and cooperation. Thus, the highest level of development was found in Brazil (97.68), followed by Mexico (81.68), Uruguay (75.15), the Dominican Republic (75.07), Chile (68.83), Costa Rica (67.46), Colombia (63.74), Peru (55.68) and Argentina (50.12). Despite the achievements made to date, the region's lag on some fronts requires it to take measures to improve its response capacity to cyber risks and threats.

The objective of the index proposed in this work is to evaluate the scope of cybersecurity disclosure, taking the main global acceptance standards as a basis for its elaboration. It is made up of 27 individual elements, associated with one of four dimensions, as follows: governance (5), strategy (6), risk management (13) and financial implications (3). The selection of the items that make up the Cybersecurity Disclosure Index (CDI) was made from previous literature, studies of international accounting and auditing firms, international standards, such as ISO and GRI, SEC guides (2011; 2018), the General Data Protection Regulation (GDPR) of the European Union, and other documents proposed by entities with wide international recognition, such as OECD, IDB, OAS and the Global Reporting Initiative (GRI), among others.

The results obtained through its application to a sample of listed companies in Latin America show that, although there has been a sustained increase in the level of disclosure of information on cybersecurity in Latin American companies, a trend evidenced in other studies [18,21,60], the average annual mark does not exceed 40% of the total mark. For this reason, we can conclude that the level of disclosure is low; this is also demonstrated by the findings obtained by [20].

Regarding the longitudinal analysis by countries, the application of the instrument allows us to verify that the country that constantly disclosed to a greater degree during the period of 2016–2020 was Argentina; according to the sample, 86% of the companies in this country reported to the SEC, and mostly belong to the financial sector, a situation that according to [22] may generate an increase in cybersecurity disclosure. The measurement of the disclosure by sector shows how the financial sector obtained leadership during the five years of the study. In this sense, we can note that in the countries under study and from the international level, regulations for this sector have increased and are more demanding, because these organizations present a greater vulnerability to cyberattacks. We agree with [20] that cybersecurity information disclosure practices have evolved in light of the expectations of financial regulators. In addition, the comparative study by dimension indicates how the strategy dimension obtained the highest score during the period of 2016–2020. Specifically, based on the results obtained, we can conclude that—in general terms—the companies under analysis revealed information about the actions designed over the long-term, such as cybersecurity strategies, and have been interested in communicating the measures related to cyber risk management, specifying how this emerging risk is linked, little by little, to other corporate risks. The disclosures also allow visualization of how the board of directors and management are gradually linked in the supervision of cybersecurity matters.

The proposed disclosure index constitutes a novel instrument, which makes it possible to assess the scope of the disclosure of information on cybersecurity. Considering the fact that cybersecurity has become a business responsibility and an important part of non-financial information, the information revealed by this index can constitute a point of reference for the assessment of organizations in this regard by the different stakeholders.

Since the proposed index has been prepared based on international standards related to business cyber risk and has not been tested, it may be applicable in future research to companies that carry out their activity in other geographical areas. In this way, it is expected that this index will become a starting point for comparative analysis in other countries in different periods of time, which will allow knowing trends in the disclosure of cybersecurity information from different perspectives.

On the other hand, we recognize that this study has some limitations; the proposed index shows a significant inclination towards the United States country context, which, according to previous literature, presents the greatest advances in the dissemination of

information on business cybersecurity. However, the specific characteristics of this environment could make it difficult to generalize to other regions of the world. As our sample is made up of listed companies listed in the main stock indices of the countries under study, our findings are relevant for this type of company.

Finally, we consider that this work will help to project future lines of research through the carrying out of other types of studies that allow us to understand why companies in Latin America voluntarily report their cybersecurity disclosures, as well as analyze what factors affect the scope of cybersecurity information disclosure by organizations or, in turn, what implications at the level of disclosure on key aspects it has for the company, such as the price of shares, or even its level of reputation in the market.

**Author Contributions:** Conceptualization, L.R.A. and M.R.; methodology, M.E.G.M. and L.R.A.; formal analysis, M.E.G.M. and M.R.; writing—original draft preparation, M.R.; writing—review and editing, L.R.A. and M.E.G.M.; writing-review and editing, V. All authors have read and agreed to the published version of the manuscript.

**Funding:** This research received no external funding.

**Institutional Review Board Statement:** Not applicable.

**Informed Consent Statement:** Not applicable.

**Data Availability Statement:** Not applicable.

**Conflicts of Interest:** The authors declare no conflict of interest.

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
