# Peer review of "The Disclosures of Information on Cybersecurity in Listed Companies in Latin America—Proposal for a Cybersecurity Disclosure Index"

_sustainability, doi:10.3390/su14031390_

Round 1

Reviewer 1 Report

The paper is ready for publication.

Author Response

The proofreading and editing of the English language of the manuscript has been carried out with the signature Proofreading Services. However, the entire document has been revised again.

Thanks

Reviewer 2 Report

The paper examines a topical issue on the cybersecurity disclosures by companies. Cybersecurity has become one of the greatest risks and threats for companies in today’s digital economy. The paper also examines this topic in the region of Latin America which has been largely neglected in terms of the cybersecurity disclosures by company. The paper also develops a disclosure index which is a novel instrument for measuring the content of disclosure on cybersecurity. I believe the paper has an important contribution to the literature and would like to highlight some issues:

  1. The use of hypothesis

From the title, it appears the paper is normative in nature as it has “a proposal for a cybersecurity disclosure index”. I have no problems with a normative paper of this nature but in the paper the authors propose hypothesis for testing (which is more positivist). The hypothesis testing section seems at odds with the title and the rest of the paper. The hypothesis in themselves don’t necessarily need to be stated as the paper illustrates that cybersecurity disclosures of companies in Latin America are increasing and there are differences in disclosures between companies. So I would suggest that possibly of removing the hypothesis testing and focusing on outlining the paper’s development of the disclosure index.

2. Limitations of the research

I’m surprised the paper does not state any limitations of the research. If the paper were to follow my suggestion above then it could state that the paper proposes a method that while hasn’t been tested is a practical instrument from industry and could be tested in future research. The paper could also discuss limitations related to focusing on one region which might have specific characteristics that would make it difficult to generalize to other regions in the world.

iii.)       Avenues for future research

The paper outlines some avenues for future research but I believe there are other areas that could be identified such as understanding why companies in Latin America are voluntarily reporting on their cybersecurity disclosures.

Other issues;

  1. The last section is section 5.Discussion and is strange because it is also the conclusion. I suggest renaming the section to Discussion and Conclusion.

Author Response

Response to Reviewer 2 Comments

The proofreading and editing of the English language of the manuscript has been carried out with the signature Proofreading Services. However, the entire document has been revised again.
Point 1. The hypothesis testing section seems to disagree with the title and the rest of the article. He suggests eliminating the hypothesis test and focusing on outlining the development of the disclosure index.
Answer 1. Following your prompts, we have eliminated hypothesis testing and focused on outlining the development of the disclosure index.
Point 2. The article does not indicate limitations of the research.
Answer 2. The following limitations are incorporated: the proposed index shows a significant inclination towards the United States context, according to the previous literature, this country presents the greatest advances in the dissemination of information on business cybersecurity. However, the specific characteristics of this environment could make it difficult to generalize some parts of the instrument to other regions of the world. Given that our sample is made up of listed companies of the main stock market indices of the countries under study, our findings are relevant for these types of companies (page 21).
Point 3. The document describes some avenues for future research, but other areas could be identified:
Answer 3. The suggested future research path is incorporated: “understand why companies in Latin America voluntarily report their cybersecurity disclosures”. Coherent theme, complement to this research (page 21).
Point 4. Change the title of section 5
Answer 4. The title change of section 5 is made and the suggested title “Discussion and conclusion” (page 19) is incorporated.

Reviewer 3 Report

The paper examines the level of mandatory and voluntary disclosure provided by a sample of Latin American companies through annual reports and 20 F forms furing the 2016-2020 period.  To this end, the authors performed a content analysis based on a disclosure index composed of 27 associated elements grouped in four dimensions: governance, strategy, risk management, and financial implications.

The paper focuses on an interesting topic. However, it needs many improvements to be considered worthy of publication

The authors should clarify what they intend for mandatory and voluntary disclosure on cybersecurity. Does cybersecurity disclosure is mandatory or voluntary in Latin American countries? If mandatory, what are the information companies must disclose? Moreover, they should clarify if the paper is built on a theoretical framework to explain cybersecurity disclosure practices.

In the paragraph “2.1. Disclosure of corporate cybersecurity in the international arena”, the authors may consider the possibility to insert a table in which they clarify the different mandatory and voluntary initiatives affecting corporate cybersecurity in the Latin American context. This would support the readers in better understanding the paper’s background.

The literature review is absent. The authors need to enhance the general discourse of risk disclosure and briefly mention previous studies investigating cybersecurity risk disclosure. I suggest authors consider, among the others, the following papers:

  • Oliveira, J., Lima Rodrigues, L. and Craig, R. (2011), “Risk-related disclosures by non-finance companies”, Managerial Auditing Journal, Vol. 26 No. 9, pp. 817-839.
  • Linsley, P.M. and Shrives, P.J. (2006), “Risk reporting: a study of risk disclosures in the annual reports of UK companies”, The British Accounting Review, Vol. 38 No. 4, pp. 387-404.
  • Linsley, P.M., Shrives, P.J. and Crumpton, M. (2006), “Risk disclosure: an exploratory study of UK and Canadian banks”, Journal of Banking Regulation, Vol. 7 Nos 3/4, pp. 268-282.
  • Guthrie, J., Rossi, F. M., Orelli, R. L., & Nicolò, G. (2020). Investigating risk disclosures in Italian integrated reports. Meditari Accountancy Research.

In the current state, hypotheses development is weak. Each hypothesis needs to be built on a solid and coherent theoretical framework and linked to empirical evidence found in previous similar studies if existing.

The discussion of results needs to be improved by including more theoretical implications and comparisons with previous similar studies.

Conclusions, limitations and practical implications are missing.

The paper needs to be proofread. Authors must carefully check for the presence of mistakes and formal errors, e.g. “due to the first guidance guide” – line 68; “In the international stock markey” line 142…

Author Response

Response to Reviewer 3 Comments

The proofreading and editing of the English language of the manuscript has been carried out with the signature Proofreading Services. However, the entire document has been revised again.
Point 1. Clarify if disclosure on cybersecurity is mandatory or voluntary in Latin American countries.
Answer 1. The following clarification is presented: the disclosure of cybersecurity information in Latin America is voluntary, however, it is mandatory to present the disclosures of risk factors regulated by the SEC in Form 20F number 3 D for listed companies (page 8)
Point 2. If it is mandatory, what information must they disclose?
Answer 2. The question is answered: the mandatory information for listed companies in Latin America is that requested by the SEC in Form 20F number 3 D for foreign issuers (page 8).
Point 3. Clarify if the document is based on a theoretical framework to explain disclosure practices on cybersecurity.
Answer 3. The following clarification is presented: our study is developed based on institutional theory, considering that the countries of the region under study present their own factors that play a fundamental role in corporate practices, giving rise to differences between countries. Thus, institutional factors condition corporate transparency, which is reflected from the disclosure of information (page 7).
Point 4. In section 2.1. Disclosure of corporate cybersecurity in the international arena, insert a table to clarify the mandatory and voluntary initiatives that affect corporate cybersecurity in the Latin American context.
Response 4. The information requested is incorporated into Table 1, presenting the guidelines that determine the disclosure of voluntary and mandatory cybersecurity information (page 8).
Point 5. Literature review.
Answer 5. Previous studies investigating cybersecurity risk disclosure are briefly mentioned. It has not been possible to do a general risk disclosure review, but an attempt has been made to cover much of the unique studies related to cybersecurity risk disclosure (page 8).
Point 6. Development of hypotheses: it is weak. Incorporate theoretical framework and link with previous studies.
Answer 6. We have responded to the suggestion of the reviewer 2, the hypothesis statement has been eliminated. However, and despite this, following his instructions, a brief reference has been made to the theoretical framework and to the preceding investigations that investigate the disclosure of cybersecurity risks (pages 7 and 8).
Point 7. Improve the discussion of the results: include theoretical implications and comparisons with previous similar studies.
Answer 7. The theoretical implications and comparisons with previous similar studies have been incorporated into the discussion of results (page 20).
Point 8. Conclusions, limitations and practical implications:
Answer 8. Conclusions have been reviewed and, following his instructions, the limitations of the study are presented (pages 20, 21).
Point 9. Review the errors
Answer 9. The formal errors have been corrected and the entire document has been revised (lines 68, 104, 105).

Round 2

Reviewer 3 Report

The paper has been improved following reviewers' comments. Now, it is more clear and consistent with the journal's standards. Accordingly, I suggest a final accept.